# Evaluation of Physical Literacy in 9- to 11-Year-Old Children: Reliability and Validity of Two Measurement Tools in Three Southeastern European Countries

**DOI:** 10.3390/children10111722

**Published:** 2023-10-24

**Authors:** Petra Rajkovic Vuletic, Marijana Geets Kesic, Barbara Gilic, Miran Pehar, Edin Uzicanin, Kemal Idrizovic, Damir Sekulic

**Affiliations:** 1Faculty of Kinesiology, University of Split, 21000 Split, Croatia; petra.rajkovic@kifst.eu (P.R.V.); markes@kifst.hr (M.G.K.); barbara.gilic@kifst.eu (B.G.); 2Faculty of Kinesiology, University of Zagreb, 10000 Zagreb, Croatia; 3Faculty of Science and Education, University of Mostar, 88000 Mostar, Bosnia and Herzegovina; miran.pehar@fpmoz.sum.ba; 4Faculty of Sport and Physical Education, University of Tuzla, 75000 Tuzla, Bosnia and Herzegovina; edin.uzicanin@untz.ba; 5Faculty for Sport and Physical Education, University of Montenegro, 81400 Niksic, Montenegro; kemo@t-com.me

**Keywords:** physical literacy, primary school, test–retest, internal consistency, criterion validity

## Abstract

The awareness of the importance of physical literacy (PL) is globally increasing; however, knowledge of the applicability of PL measurement tools in southeastern Europe is limited. The aim of this study was to evaluate the reliability and validity of translated versions of the CAPL-2 and PLAYself questionnaires in 9- to 11-year-old elementary school children from Croatia, Bosnia and Herzegovina, and Montenegro. The participants were 303 children (141 girls; all 9 to 11 years of age) from Croatia (*n* = 71), Bosnia and Herzegovina (*n* = 162), and Montenegro (*n* = 70), enrolled in regular elementary school. The participants were tested throughout a test–retest procedure using two PL evaluation tools, i.e., the Canadian Assessment of Physical Literacy (shorter version, CAPL-2) and the Physical Literacy Assessment of Youth (PLAYself) questionnaires. With an intraclass correlation (ICC) of 0.70–0.80 for specific questionnaire subscales and 0.84 for the total score, PLAYself was found to be reliable. With Kappa values of 0.11–0.23 and a percentage of absolute agreement of less than 62%, CAPL-2 appeared to be less reliable. Factors related to sport participation were significantly positively associated with the PLAYself score, indicating its proper validity. In conclusion, we suggest the usage of the PLAYself questionnaire in further studies examining PL in children of a similar age in the region. Future studies in other age groups and languages are also warranted.

## 1. Introduction

Although the benefits of physical activity (PA) are well known, people, including children, are insufficiently physically active [1]. Current research clearly indicates that PA decreases with age, especially in adolescence [2,3,4]. Of special concern is the fact that recent reviews highlighted a progressive decline in moderate- and high-intensity PA even before adolescence, starting from early and middle childhood [5]. Consequently, the interest in strategies that could positively influence the overall PA level in children and youth and in the concept of physical literacy (PL) has increased [6,7]. PL can be defined as a “disposition to capitalize on our human-embodied capability wherein the individual has the motivation, confidence, physical competence, knowledge, and understanding to value and take responsibility for maintaining purposeful physical pursuits and activities throughout the life course” [8]. In other words, it is expected that ”physically-literate-individuals“ will be more likely to achieve a ”physically active way of life“ [9]. Consequently, it is of utmost importance to identify individuals with low levels of PL in order to educate them about the necessity of PA and on how to achieve appropriate levels of PA. To do this, it is necessary to identify the most appropriate ways of evaluating PL in different sub-populations.

So far, various strategies to evaluate PL have been developed, but there is no clear consensus on the most appropriate one [10,11,12,13]. The most popular PL evaluation tools are the (i) Canadian Assessment of Physical Literacy (CAPL) [14] and the (ii) Physical Literacy Assessment of Youth (PLAY) [10] protocols. The CAPL protocol (as well as its shorter version CAPL-2) consists of a battery of standardized evaluation protocols that have been regularly proven to be valid and reliable for PL evaluation in children and adolescents [15,16]. The CAPL-2 consists of three protocols aimed at the evaluation of physical competence, two protocols focused on daily behavior, and a questionnaire of 22 items used to evaluate knowledge, understanding, motivation, and self-confidence during physical exercising. The PLAY evaluation protocol consists of six tools; namely, PLAYfun, PLAYbasic, PLAYself, PLAYparent, PLAYcoach, and PLAYinventory, which use workbooks, forms, and score sheets to achieve a specific measurement of PL in children and youth [17]. As for the CAPL, the PLAY tools are also proven to be reliable and valid for the evaluation of PL in children and youth [18,19].

The concept of PL is multidimensional, and scientists share the opinion that each component of PL should be evaluated. However, this is challenging, as it requires significant resources and time. Therefore, most studies have used standardized questionnaires that partially define PL and/or some of its features. The most used questionnaires are those included in CAPL-2 and PLAY [17,20]. The CAPL questionnaires examine the cognitive domain of PL (knowledge and understanding) through questions aimed to evaluate (i) the understanding of the recommendations regarding PA and sedentary behavior, (ii) the knowledge of overall fitness parameters and movement and how to improve them, and (iii) the different perceptions of health [13,21]. On the other hand, the PLAYself questionnaires include items aimed to access (i) confidence and motivation, (ii) knowledge and understanding, and (iii) environmental engagement ability [19].

From this literature overview, it is clear that the concept of PL deserves special attention, particularly in children and youth. Meanwhile, CAPL-2, and PLAYself questionnaires were repeatedly found to be reliable and valid measurement tools [19,21,22,23,24,25]. However, the reliability and validity of the PL-measurement tools in southeastern Europe is rarely evidenced, and only one study examined this issue in adolescents [26], while: (i) there is an evident lack of research examining this issue in younger children, and (ii) to the best of our knowledge, no study has been conducted on elementary school children in southeastern Europe (the territory where similar Slavic languages are spoken) [27]. Therefore, the aim of this study was to evaluate the reliability and validity of translated versions of the CAPL-2 and PLAYself questionnaires in 9- to 11-year-old children from Croatia, Bosnia and Herzegovina, and Montenegro. We hypothesized that both questionnaires would be reliable and valid tools for the evaluation of PL among the studied participants. Considering the similarity of the languages spoken in the examined region, we believe that the study findings will be applicable in a wider context and in other surrounding countries

## 2. Materials and Methods

### 2.1. Participants

The participants in this study were 303 children (141 girls), elementary school students from Croatia (*n* = 71), Bosnia and Herzegovina (*n* = 162), and Montenegro (*n* = 70). At the moment of testing, all children were between 9 and 11 years of age and attended the 3rd or 4th grade of the regular school program. Their parents (legal guardians) were informed about the study protocol and aims and provided their written consent for the participation of the children in the investigation. The study was approved by the Ethical Committee of the University of Split, Faculty of Kinesiology (EBO: 2181-205-02-01-21-0011; date of approval, 23 September 2021).

### 2.2. Variables and Testing

The variables in this study were school grade (3rd or 4th grade; younger vs. older school age), gender (male and female), age (in years), sport factors, CAPL-2 and PLAYself questionnaires.

The evaluation of the sport factors involved questions on current sport participation (yes or no), experience in sport (in years, i.e., one, two, three, four, five, etc.), and the number of training sessions per week (one, two, three, etc.).

The CAPL-2 knowledge and understanding questionnaire included 12 questions aimed at evaluating the participants’ knowledge of the necessity and importance of daily PA, the problem of sedentarism, the importance and definition of cardiorespiratory fitness and muscular strength, the concept of fitness and its impact on a physically active lifestyle, ways of improving one’s motor skills, and the components of overall fitness. Here are some examples of the questions we asked: “There are many different kinds of fitness, for example, endurance fitness, aerobic fitness, cardiorespiratory fitness. What is cardiorespiratory fitness?“; “if you wanted to improve a sport skill (like kicking and catching a ball), what would be the best type of training?” the participants had to choose the correct answer from four options; each correct answer was assigned 1 point, while each incorrect answer received a score of 0, resulting in a total score from 0 to 12.

The PLAYself questionnaire is used in children and adolescents in order to establish their perceived level of PL and includes four subsections: (i) environment, assessing the degree of movement confidence in different environments (i.e., gym, water, snow); (ii) affective and cognitive aspects related to PL that determine individuals’ self-efficacy with respect to their participation in PA through items such as “It does not take me long to learn new skills, sports, or activities”, “I think that being active is important for my health and well-being”, and “I understand the words that coaches and PE teachers use”; (iii) relative ranking of literacy, numeracy, and physical literacy in different settings including school, home, and social life with friends, examining how much an individual values each literacy type; and (iv) fitness, whose evaluation is determined by the score assigned to the item “My fitness is good enough to let me perform all the activities I choose”, which is not included in the final score. The final score consists of the subtotals from the first three subsections divided by the number of questions (27 in total). The maximum PLAYself score is 100, representing a high self-perceived PL.

The children were tested during school hours using the test–retest procedure, with 5–7 days between the testing sessions. Online questionnaire forms using the SurveyMonkey platform were prepared and answered in computer classrooms. The questionnaires were anonymized, and no personal data were asked. However, for the purpose of pairing the test and retest results (please see later for statistical analyses), the children were asked to use a self-selected code in both testing sessions, which they should send by SMS/WhatsApp to one of their parents (in case it was forgotten). One of the examiners was available if the children needed help during the test, but the examiner was positioned behind the computer screen and could not see the answers. Testing was carried out in small groups, up to 10 children, and the children could not see the other participants’ answers. PLAYself and CAPL-2 questionnaires were previously proven to be feasible, valid, and reliable for 14- to 18-year-old Croatian adolescents [26,28]. However, knowing the similarity of the Croatian, Bosnian and Herzegovinian, and Montenegrin languages (other than some grammar and punctuation differences, these three languages are practically equal), only some minor corrections and adaptations were applied for conducting the tests in Bosnia and Herzegovina, and Montenegro.

### 2.3. Statistical Analyses

The statistical analyses were performed in several phases. In the first phase, the test–retest reliability of PLAYself was established by calculating intraclass correlation coefficients (ICC) with 95% confidence intervals (95%CI), using a two-way mixed-effect model with absolute agreement. The ICC values were interpreted as poor if <0.5, moderate if between 0.5 and 0.75, good if between 0.75 and 0.9, and excellent if >0.90 [29]. Cronbach’s alpha coefficients (CA) were calculated to define the internal consistency of the test and retest. The CA values were interpreted as previously suggested, i.e., as unacceptable if <0.5, poor if ≥0.5 and <0.60, questionable if ≥0.60 and <0.7, acceptable if ≥0.70 and <0.8, good if ≥0.80 and <0.9, and excellent if ≥0.90 [30]. In brief, the CA is an indicator of the correlation of items, and therefore, a high CA value in general justifies the summarizing of the related items. To evaluate the test–retest reliability of the CAPL-2, somewhat different statistical analyses were conducted, because the responses in this questionnaire were generally dichotomous. Consequently, for each question, weighted Cohen’s Kappa coefficients (Cκ) with 95%CI and percent of the overall agreement (p0) were calculated. The Cκ values were interpreted as follows: slight = 0.00–0.20, fair = 0.21–0.40, moderate = 0.41–0.60, substantial = 0.61–0.80, and almost perfect = 0.81–1.00. A p0 ≥ 80% was considered acceptable [31].

Following the calculation of the reliability, the evaluation of the test validity comprised several analyses. Firstly, construct validity was evaluated by principal component factor analysis, using the Gutman–Kaiser criterion of extraction with a varimax raw rotation. The discriminative validity of CAPL-2 and PLAYself were determined by a *t*-test for independent samples, calculating the differences between (i) boys and girls, (ii) older and younger children (according to their school grade), and (iii) children involved in sports and children non-involved in sports. Additionally, Cohen’s d as effect size (ES) with 95% confidence intervals (95%CI) were calculated and interpreted for significant *t*-test differences, with ≥0.2 as small ES, ≥0.5 as medium ES, and ≥0.8 as large ES. Also, we calculated the Spearman’s rank order correlations to evidence the association between the PL scores of time dedicated to sport activities and number of training session per week, which allowed us to additionally evaluate the criterion-related validity of the applied questionnaires. The Pearson’s r values were interpreted as very weak if in the range of 0.00–0.19, weak if in the range of 0.20–0.39, moderate if in the range of 0.40–0.69, strong if in the range of 0.70–0.89, and very strong if in the range of 0.90–1.00 [32].

The statistical significance of 95% (*p* = 0.05) was applied, and the statistical analyses were performed by Statistica version13 (Palo Alto, California, CA, USA).

## 3. Results

In general, PLAYself showed acceptable internal consistency. While the “literacy” and “numeracy” subsections presented somewhat low α values (0.55–0.60), other subsections had higher alpha values, indicating acceptable-to-good internal consistency of the questionnaire for early-school-age children. With values ranging between 0.70 and 0.84, the ICC values indicated good test–retest reliability (Table 1).

The calculated Cκ values and percent of absolute agreement of the test and retest responses showed a low reliability for the CAPL-2 questionnaire. In brief, none of the Cκ values exceeded 0.23 (interpreted as “fair”), while the percent of absolute agreement was generally low, reaching a maximum of 61.1% (Table 2).

On the basis of the Gutman–Kaiser criterion (e.g., variance of the factor > 1), the factor analysis extracted two significant latent dimensions of the PLAYself questionnaire. The first latent dimension (F1) explained 44% of the variance, being saturated with the variance of the last three subscales. The second latent dimension was correlated to the first two subscales and explained 33% of the system variance (Table 3).

The descriptive statistics and *t*-test differences between boys and girls in the PLAYself subscores (Figure 1A) and total score (Figure 1B) evidenced higher scores for girls in literacy-subscale (*t*-test = 3.11, *p* < 0.05; ES (95%CI): 0.63 (0.18–1.07)—medium ES), numeracy-subscale (*t*-test = 3.14, *p* < 0.05; ES (95%CI): 0.67 (0.22–1.18)—medium ES), and physical literacy subscale (*t*-test = 2.89, *p* < 0.05; ES (95%CI): 0.59 (0.14–1.03)—medium ES), and no significant difference between boys and girls in the PLAYself total score (*t*-test = 0.31, *p* > 0.05).

When school ages were compared in relation to the PLAYself subscores (Figure 2A) and PLAYself total score (Figure 2B), no significant differences were found between younger and older children

The children involved in sport (Athletes) achieved higher scores in environment- (*t*-test = 8.14, *p* < 0.001; ES (95%CI): 1.23 (0.76–1.71)—large ES), and self-description- (*t*-test = 3.14, *p* < 0.01; ES (95%CI): 0.73 (0.28–1.18)—medium ES) -subscales of PLAYself (Environment and Self-description) than their peers who were not involved in sports (Nonathletes). Nonathletes reported higher scores in the literacy-subscore (*t*-test = 3.14, *p* < 0.05; ES (95%CI): 0.66 (0.21–1.10)—medium ES) and numeracy-subscore (*t*-test = 3.13, *p* < 0.05; ES (95%CI): 0.64 (0.19–1.09)—medium ES) (Figure 3A). Generally, the Athletes obtained a higher PLAYself total score, indicating the proper validity of the PLAYSself questionnaire (*t*-test = 3.15, *p* < 0.05; ES (95%CI): 0.68 (0.23–1.12)—medium ES) (Figure 3B)

The children who reported a longer involvement in sport achieved higher PLAYself scores (Pearson’s correlation = 0.42, *p* < 0.01; moderate correlation) than those who had recently started to practice sport, and a higher number of training sessions per week was also positively correlated with the PLAYself score (Pearson’s correlation = 0.43, *p* < 0.01; moderate correlation). These results support the positive association between involvement in sports and PL.

## 4. Discussion

This study aimed to evaluate the reliability and validity of the translated versions of the tools aimed at the evaluation of PL in children aged 9 to 11 years. The most important findings of our research are as follows. First, we evidenced the proper reliability of the translated version of the PLAYself questionnaire and the insufficient reliability of the CAPL-2 in three studied southeastern European countries. Second, since sport participation was positively associated with the PLAYself total score, PLAYself showed good validity for 9- to 11-year-old children. Initially, we hypothesized the proper reliability and validity of both PL questionnaires, but our findings indicated that the initial study hypothesis could be only partially accepted.

### 4.1. Reliability

With good internal consistency and test–retest correlation, the translated version of PLAYself showed appropriate stability of the measurements and, therefore, was confirmed as a reliable tool for the evaluation of PL in early-school-age children. This could be due to several factors. Firstly, PLAYself is a questionnaire designed to assess participation in different environments, the affective/cognitive domains of physical literacy, and the valuing of the different types of literacy [17,19]. The questions included in this measurement tool seemed to be clear and understandable for 9- to 11-year-old children, which is the main prerequisite of testing reliability. Also, in the current investigation, follow-up testing was carried out after 5–7 days. During such a short period, the children most likely did not change their personal opinions on the tested issue. Consequently, the results remained consistent, which assured a high reliability.

At the same time, the translated version of the CAPL-2 did not result as reliable. A possible reason for this could be that the CAPL-2 is a questionnaire designed to test knowledge and understanding by focusing on the theoretical knowledge of PL and its importance. Despite the short period between the baseline and the follow-up tests, it is possible that the children learned something new about the examined topics. This resulted in dissimilarities between the baseline and the follow-up tests, decreasing the correlation between test and retest, which consequently altered the reliability parameters. Also, it is possible that the questions in the CAPL-2 were too difficult to understand or answer for 9- to 11-year-old children. As a result, the children were not able to find the correct answers and “guessed” them, which additionally reduced the consistency of the results [33].

Some studies already examined the reliability of the CAPL-2 and PLAYself questionnaires in children, and mostly confirmed the proper reliability of both measurement tools. For example, Jefferies et al. investigated 8- to 14-year-old Canadian children and revealed a moderate reliability of the PLAYself questionnaire, while another study investigating 8- to 14-year-old Canadian children demonstrated a strong inter-reliability of the PLAYself questionnaire [18,19]. Analyzing the knowledge and understanding of PL among 8- to 12-year-old Canadian children, Longmuir et al. (2018) confirmed the reliability of CAPL-2 as well [21]. However, these studies rarely simultaneously evaluated the reliability of different questionnaires. In one of these few studies investigating 14- to 18-year-old Croatian adolescents, Šunda found that both PLAYself and CAPL-2 were reliable, but PLAYself showed higher reliability and suggested that these differences in reliability could be due to specific features of the physical education curriculum in Croatia [26]. The physical education curricula in the three studied countries are mostly oriented toward the development of motor skills and conditioning capacities, while the (theoretical) knowledge and understanding of the importance of PA are poorly considered [26]. If this was the case for older participants (14-to-18 years old), it is certainly even truer for younger children, resulting in the previously discussed issues and in the poor reliability of the CAPL-2 for 3rd and 4th grade elementary school children.

### 4.2. Validity

A test is valid only if it can distinguish the groups under study [34,35]. To evaluate the validity of the questionnaires assessing PL, differences by gender, age, and sport participation (i.e., involved vs. not involved in sports) were analyzed. In regard to gender differences, the PLAYself results for boys and girls did not differ. Since PLAYself is designed to carry out assessments within a population group (i.e., within participants of the same gender, not between groups of different genders), such results are not surprising. First, physical education in Croatia, Bosnia and Herzegovina, and Montenegro takes place in mixed classes, and the school systems in the three countries apply standardized norms for monitoring, checking, and evaluating the kinanthropological characteristics of the students by gender [36]. Therefore, comparisons are typically carried out among girls or among boys. As the students do not compare themselves to those of the opposite gender, differences between boys and girls are actually not considered nor expected. Indeed, such results are consistent with previous studies confirming no gender differences in PL measures between boys and girls [18,20,28,37,38].

We did not find differences in PLAYself between 3rd- and 4th-grade students. Although studies rarely directly compared age groups in PL evaluations as we did herein, but mostly correlated age with PL, we can say that the results of previous studies are equivocal. For example, our results are in contrast with those of a Canadian study reporting that PL increases with age [21], but support a study investigating 14-to-18-year-old Croatian adolescents which revealed no to trivial associations between age and PL identified by PLAYself and CAPL-2, respectively [37]. Although age could be expected to be correlated with PL, we believe that our findings (i.e., no difference in PL between the studied age categories) are somewhat specific. Firstly, as said previously, the physical-education systems in the studied countries are mostly focused on the development of motor abilities (i.e., motor and functional abilities and motor knowledge), while the development of an overall theoretical knowledge of PL (i.e., importance in everyday life, benefits for improving and preserving health, lifestyle habits, etc.) is negligible [37]. This was especially evident for the children in the first four grades of elementary school, as physical education classes are taught by elementary school teachers and not by physical education teachers [39]. Secondly, it is most likely that the previously mentioned specificities of PLAYself (i.e., assessment within a population group) also had an impact on our results. Possibly, 3rd -grade students compared themselves to other 3rd -grade students, while 4th-grade students compared themselves to other 4th-grade students, which resulted in no association between age and PA.

The validity of the PL measurements was additionally analyzed according to the level of sport participation. Our results revealed differences in the PLAYself scores between students participating and students not participating in sport. Previous studies regularly reported that sport participation is directly linked to a higher PL [19,37,38]. Therefore, it can be concluded that the translated version of the PLAYself questionnaire has appropriate validity for 3rd- and 4th-grade students. The positive correlations of PLAYself with “experience in sport” and “number of training sessions per week” examined in the current study additionally support such conclusions. Specifically, given that PLAYself is a questionnaire that assesses one’s abilities and self-efficacy in practicing PA, physical exercise, and sports, it was reasonable to expect that the children who practiced sport activities for longer times and more frequently would report a better PL. Since the evaluation of self-efficacy involves the judgment of one’s own ability to organize and perform certain tasks [40], it was logical that facets of sport participation would be positively associated with PL. These considerations are supported by previous studies showing that sports participation is positively related to self-efficacy among 13- to 17- year-old students [41].

### 4.3. Limitations and Strengths of the Study

The main limitation of this study is the evaluation of PL using questionnaires that assessed only the cognitive (knowledge and understanding) and affective (confidence and motivation) domains of PA. This was due to the fact that the concept of PL in the studied countries is still developing. We did not evaluate other PL variables such as fitness status to further determine the validity of the questionnaire, mostly because of the differences in fitness evaluation standards in the three countries. However, as studies investigating elementary school children are lacking, we decided to carry out a preliminary assessment of PL in this group to establish a foundation for further research on the physical education concept related to PL and its development in the elementary school.

Meanwhile, this is one of the first studies in Europe, and possibly the first in southeastern Europe, investigating PL among children of young school ages. This is also one of the rare studies that simultaneously investigated the reliability and the validity of two PL assessment tools, allowing an objective comparison between them. Although we studied versions of the PL evaluation tools in only three countries from the territory of former Yugoslavia, where similar Slavic languages are spoken, we hope that our results will allow a wider usage of these measurement tools.

## 5. Conclusions

We confirmed good reliability of the translated version of the PLAYself questionnaires for children aged 9–11 years. In contrast, the reliability of the CAPL-2 was not confirmed. Therefore, we suggest the usage of PLAYself in further studies examining PL in children of similar ages in the region.

The validity of PLAYself was found to be appropriate. This conclusion is based on (i) the significant differences that we observed between the children involved in sports and those not involved (with better PL for those who participated in sports) and (ii) the significant correlation of the PLAYself scores relative to the time of the sport involvement and training frequency.

While this study involved specific groups of participants (school-age children) from urban regions, further analyses are needed in other age groups and environments. In addition, intervention studies aimed at finding the most effective methods to improve PL in school-age children are warranted.

## Figures and Tables

**Figure 1 children-10-01722-f001:**
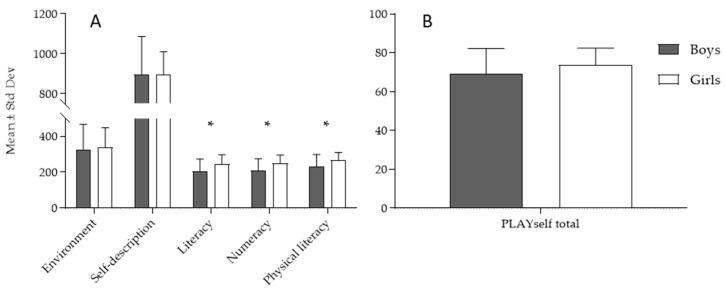
Descriptive statistics and significance of the *t*-test differences (* denotes significance with *p* < 0.05) between boys and girls in the PLAYself subscores (**A**) and total score (**B**).

**Figure 2 children-10-01722-f002:**
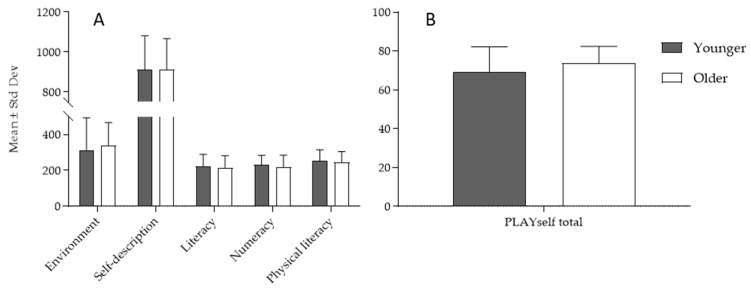
Descriptive statistics in younger and older children in the PLAYself subscores (**A**) and total score (**B**).

**Figure 3 children-10-01722-f003:**
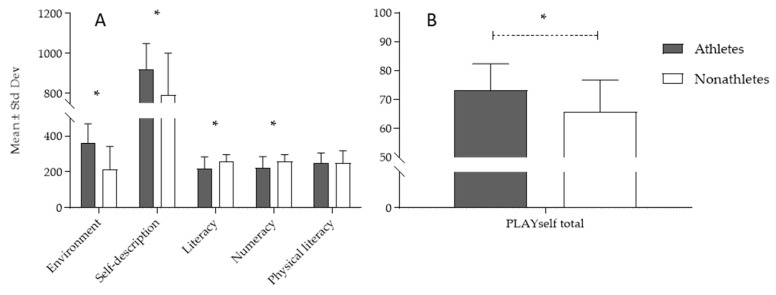
Descriptive statistics and significance of the *t*-test differences (* denotes significance with *p* < 0.05) between children participating in sports (Athletes) and those not involved in sports (Nonathletes) in the PLAYself subscores (**A**) and total score (**B**).

**Table 1 children-10-01722-t001:** Test–retest reliability and internal consistency of the PLAYself questionnaire (α—Cronbach Alpha, ICC—Test–retest intraclass correlation coefficient, CI—confidence interval).

	Test	Retest	Reliability
Variable/Subscale	α	α	ICC (95%CI)
Environment	0.72	0.81	0.80 (0.75–0.85)
Self-description	0.74	0.77	0.77 (0.70–0.84)
Literacy	0.59	0.60	0.72 (0.64–0.82)
Numeracy	0.55	0.57	0.70 (0.61–0.79)
Physical literacy	0.78	0.79	0.78 (0.71–0.87)
Total score			0.84 (0.75–0.93)

**Table 2 children-10-01722-t002:** Test–retest reliability analysis for the CAPL-2 questionnaire (Cκ—Cohen’s kappa coefficient; 95%CI—95% confidence interval, p0—percent of absolute agreement/test retest percentage of the equally responded queries).

Item	Cκ	p0
Q1	0.23 (0.13–0.33)	56.6
Q2	0.18 (0.08–0.28)	45
Q3	0.17 (0.07–0.28)	54
Q4	0.20 (0.10–0.30)	61.1
Q5	0.11 (0.01–0.21)	35.6
Q6	0.12 (0.03–0.23)	33.2
Q7	0.15 (0.05–0.26)	47.9
Q8	0.13 (0.03–0.33)	36.6
Q9	0.14 (0.04–0.25)	28.4
Q10	0.17 (0.07–0.28)	59.3
Q11	0.22 (0.12–0.33)	73.9
Q12	0.18 (0.08–0.28)	47.7

**Table 3 children-10-01722-t003:** Factor analysis with varimax rotation of the PLAYself questionnaire.

	F1	F2
Environment	−0.06	−0.89
Self-description	0.12	−0.88
Literacy	0.93	0.08
Numeracy	0.93	0.07
Physical literacy	0.66	−0.29
**Factor variance**	**2.19**	**1.65**
**Explained variance**	**44%**	**33%**

## Data Availability

Data are available upon reasonable request.

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
