# Peer review of "Evaluation of Physical Literacy in 9- to 11-Year-Old Children: Reliability and Validity of Two Measurement Tools in Three Southeastern European Countries"

_children, 2023, doi:10.3390/children10111722_

Round 1

Reviewer 1 Report

Reviewer’s comments to author

The present study was designed to assess to evaluate the reliability and validity of translated versions of the CAPL-2 and PLAY questionnaires in 9- to 11-year-old children from Croatia, Bosnia and Herzegovina, and Montenegro. To my opinion, researchers here followed an appropriate methodology to investigate the above research questions. For these reasons, the present study may represent an important addition to the existing literature. Therefore, I believe it is publishable in its current form with a minor revision. Below, I offer both general and specific comments regarding the current manuscript and, where appropriate, I have made suggestions to the authors regarding potential strategies for addressing these tasks.

General Comments:

·       An innovative topic

·       It covers the aims and the scope of this scientific journal

·       It follows an appropriate methodology to investigate the research questions

·       A very well written and easy to read text

Moreover,

·       In general, please follow the format style of the Children journal (MDPI).

Specific comments

1. Introduction

·       Page 2, lines 46-47: This phrase her might need a reference.

·       Page 2, line 54:Canadian Assessment of Physical Literacy” (CAPL) – Please enter a reference here for the CAPL

·       Page 2, line 55: (Jean de Dieu i sur., 2021) – Please replace this author name with a reference number

·       Page 2, line 72: Please replace here physical activity with PA

2. Materials and Methods

·       Page 3, lines 103-104: The study was approved by the Ethical Committee of the University of Split, Faculty of Kinesiology. If possible, please enter here the reference number – protocol number & the acceptance date.

·       Page 3, line 113: Please replace here physical activity with PA

·       Page 4, line 157: Please add a space – gap between 95% & CI

·       Page 4, line 161: Unacceptable – Please correct it as unacceptable (no capital letters)

·       Page 4, line 162: and <0.7a – Please correct this phrase – Perhaps the correct here is “and <0.7,”

·       Page 4, line 182: Pearson’s R – Please replace R with r. Previously you mentioned that you have used the Spearman’s rank order correlation (line 178-179), but in line 182 and also in your results, you present the Pearson’s r correlation index. What is the correct?

·       Page 4, line 186: ver.13 – Please change this as version 13 or v13

3. Results

·       Page 5, line 198: Please add a dot (.) after the word … Interval)

·       Page 5, Table 1: Please add a space – gap between 95% & CI

·       Page 5, line 207: Please add a space – gap between 95% & CI

·       Page 5, line 208: Please add a dot (.) after the word … queries)

·       Page 5, line 212: Please add a dot (.) after the word questionnaire

·       Page 6, line 217: Please add a dot (.) after the word questionnaire

·       Page 6, line 229: Please add a dot (.) after the word (1B)

·       Page 7, line 238: Please add a dot (.) after the word (1B)

·       Page 7, line 249: Please add a dot (.) after the word (1B)

·       Please add the independent samples t-test values (t, p) in your results. Where there is a significant difference, please add an effect size (e.g., Cohen’s d)

4. Discussion

·       Page 8, lines 260-262: The most important findings of our research are as follows. First, we evidenced the proper reliability of the PLAYself questionnaire and the insufficient reliability of the CAPL-2. Please add here that this finding exists only for three south-eastern European countries (Croatia, Bosnia and Herzegovina, and Montenegro).  

·       Page 8, line 262: Please add a space here between these two words questionnaire,and … & delete also the comma

·       Page 8, line 291: Please replace the author’s name (Jefferies et al.) with a reference number [17]. Please do the same for the other author names appearing in the discussion section (e.g., Longmuir et al., 2018; Šunda)

·       Page 8, line 303: Please replace here physical activity with PA

·       Page 9, line 313: Please replace the word PLAYSself with PLAYself

·       Page 9, line 336: Please replace the author’s name (Šunda, 2022) with a reference number [25].

·       Page 9, lines 352-353: Please replace here physical activity with PA

·       Page 9, line 359: Please add a dot (.) after [39].

References

·        Regarding reference section, please follow the format style of the Children’s journal (MDPI).

·        In many references, the doi is missing. Please enter the dois.

·        In many references (e.g., refs’ number 9, 14, 18, 19, 26), the dois are not in line - order with the other references (they protrude - stick out of the line). Please correct them.

·       Page 11, line 427: Promotion—An – Please correct this reference here.

Reviewer’s Decision: Accepted with minor revision

Author Response

1st Reviewer

The present study was designed to assess to evaluate the reliability and validity of translated versions of the CAPL-2 and PLAY questionnaires in 9- to 11-year-old children from Croatia, Bosnia and Herzegovina, and Montenegro. To my opinion, researchers here followed an appropriate methodology to investigate the above research questions. For these reasons, the present study may represent an important addition to the existing literature. Therefore, I believe it is publishable in its current form with a minor revision. Below, I offer both general and specific comments regarding the current manuscript and, where appropriate, I have made suggestions to the authors regarding potential strategies for addressing these tasks.

Response: Dear Sir/Madam. Thank you for recognizing the potential of our work. Also, thank you for providing us with comments and suggestions which allowed us to improve the quality of the manuscript. We tried to follow it strictly and amended the manuscript accordingly. All changes are indicated in the enclosed manuscript by „track changes“ tool. With regard to specific changes, please see following text under “RESPONSES”

General Comments:

  • An innovative topic
  • It covers the aims and the scope of this scientific journal
  • It follows an appropriate methodology to investigate the research questions
  • A very well written and easy to read text

Response: Thank you once again.

Moreover,

  • In general, please follow the format style of the Children journal (MDPI).

RESPONSE: The amended manuscript we strictly followed format and style of the Journal as indicated in the instructions to authors and in template

Specific comments

  1. Introduction
  • Page 2, lines 46-47: This phrase her might need a reference.

RESPONSE: Thank you, the reference is added as suggested ( Lounsbery, M. A. F., & McKenzie, T. L. (2015). Physically literate and physically educated: A rose by any other name? Journal of Sport and Health Science, 4(2), 139-144. https://doi.org/10.1016/j.jshs.2015.02.002)

  • Page 2, line 54: “Canadian Assessment of Physical Literacy” (CAPL) – Please enter a reference here for the CAPL

RESPONSE: Reference is added as suggested, and text reads “The most popular PL evaluation tools are the (i) “Canadian Assessment of Physical Literacy“ (CAPL) (Longmuir, Gunnell, et al., 2018)  and the (ii) “Physical Literacy Assessment of Youth“ (PLAY) (de Dieu & Zhou, 2021)”

  • Page 2, line 55: (Jean de Dieu i sur., 2021) – Please replace this author name with a reference number

RESPONSE: We are sorry for this mistake, it is now replaced with number.

  • Page 2, line 72: Please replace here physical activity with PA

RESPONSE: It is now replaced, thank you.

  1. Materials and Methods
  • Page 3, lines 103-104: The study was approved by the Ethical Committee of the University of Split, Faculty of Kinesiology. If possible, please enter here the reference number – protocol number & the acceptance date.

RESPONSE: We provided details on ethical approval. Text reads: „The study was approved by the Ethical Board of the University of Split, Faculty of Kinesiology (EBO: 2181-205-02-01-21-0011; date of approval, 23 September 2021).“ (please see Methods subsection)

  • Page 3, line 113: Please replace here physical activity with PA

RESPONSE: It is now replaced.

  • Page 4, line 157: Please add a space – gap between 95% & CI

RESPONSE: It is now added.

  • Page 4, line 161: Unacceptable – Pease correct it as unacceptable (no capital letters)

RESPONSE: It is now corrected, thank you!

  • Page 4, line 162: and <0.7a – Please correct this phrase – Perhaps the correct here is “and <0.7,”

RESPONSE: It is now corrected, thank you

  • Page 4, line 182: Pearson’s R – Please replace R with r. Previously you mentioned that you have used the Spearman’s rank order correlation (line 178-179), but in line 182 and also in your results, you present the Pearson’s r correlation index. What is the correct?
  • Page 4, line 186: ver.13 – Please change this as version 13 or v13

RESPONSE: It is now changed.

  1. Results
  • Page 5, line 198: Please add a dot (.) after the word … Interval)

RESPONSE: It is now added.

  • Page 5, Table 1: Please add a space – gap between 95% & CI

RESPONSE: It is now added.

  • Page 5, line 207: Please add a space – gap between 95% & CI

RESPONSE: It is now added.

  • Page 5, line 208: Please add a dot (.) after the word … queries)

RESPONSE: It is now added.

  • Page 5, line 212: Please add a dot (.) after the word … questionnaire

RESPONSE: It is now added.

  • Page 6, line 217: Please add a dot (.) after the word … questionnaire

RESPONSE: It is now added.

  • Page 6, line 229: Please add a dot (.) after the word … (1B)

RESPONSE: It is now added.

  • Page 7, line 238: Please add a dot (.) after the word … (1B)

RESPONSE: It is now added.

  • Page 7, line 249: Please add a dot (.) after the word … (1B)

RESPONSE: It is now added.

  • Please add the independent samples t-test values (t, p) in your results. Where there is a significant difference, please add an effect size (e.g., Cohen’s d)

RESPONSE: Following your suggestion we calculated ES differences and reported it when t-test was significant.

Text in Methods (Statistics) now reads: “Additionally, Cohen’s d as effect size (ES) with 95% Confidence Intervals (95%CI) were calculated and interpreted for significant t-test differences, with ≥0.2 as small ES, ≥ 0.5 as medium ES, and ≥ 0.8 as large ES)”

Text in Results is also amended accordingly and now reads:

“The descriptive statistics and t-test differences between boys and girls in the PLAYself subscores (Figure 1A) and total score (Figure 1B) evidenced higher scores for girls in liter-acy-subscale (t-test = 3.11, p < 0.05; ES (95%CI): 0.63 (0.18-1.07) – medium ES), numeracy-subscale (t-test = 3.14, p < 0.05; ES (95%CI): 0.67 (0.22-1.18) – medium ES), and physical literacy subscale (t-test = 2.89, p < 0.05; ES (95%CI): 0.59 (0.14-1.03) – medium ES), and no significant difference between boys and girls in the PLAYself total score (t-test = 0.31, p > 0.05)” 

“The children involved in sport (Athletes) achieved higher scores in environment- (t-test = 8.14, p < 0.001; ES (95%CI): 1.23 (0.76-1.71) – large ES), and self-description- (t-test = 3.14, p < 0.01; ES (95%CI): 0.73 (0.28-1.18) – medium ES) -subscales of PLAYself (environment and self-description) than their peers who were not involved in sports (Non-athletes). Nonathletes reported higher scores in the literacy-subscore (t-test = 3.14, p < 0.05; ES (95%CI): 0.66 (0.21-1.10) – medium ES) and numeracy-subscore (t-test = 3.13, p < 0.05; ES (95%CI): 0.64 (0.19-1.09) – medium ES) (Figure 3A). Generally, the Athletes obtained a higher PLAYself total score, indicating the proper validity of the PLAYSself questionnaire (t-test = 3.15, p < 0.05; ES (95%CI): 0.68 (0.23-1.12) – medium ES) (Figure 3B)”

Please see changed text in Results subsection, thank you!

  1. Discussion
  • Page 8, lines 260-262: The most important findings of our research are as follows. First, we evidenced the proper reliability of the PLAYself questionnaire and the insufficient reliability of the CAPL-2. Please add here that this finding exists only for three south-eastern European countries (Croatia, Bosnia and Herzegovina, and Montenegro).

RESPONSE: Amended accordingly. Text now reads: „First, we evidenced the proper reliability of the translated version of the PLAYself questionnaire ,and the insufficient reliability of the CAPL-2 in three studied south-eastern European countries.“ (please see first paragraph of the Discussion section).

  • Page 8, line 262: Please add a space here between these two words questionnaire,and … & delete also the comma

RESPONSE: It is now added and deleted.

  • Page 8, line 291: Please replace the author’s name (Jefferies et al.) with a reference number [17]. Please do the same for the other author names appearing in the discussion section (e.g., Longmuir et al., 2018; Šunda)
  • Page 8, line 303: Please replace here physical activity with PA

RESPONSE: Please accept our apology for these mistakes, we chacked the whole manuscript and corrected it. Thank you!

  • Page 9, line 313: Please replace the word PLAYSself with PLAYself

RESPONSE: It is now replaced.

  • Page 9, line 336: Please replace the author’s name (Šunda, 2022) with a reference number [25].

RESPONSE: It is now replaced.

  • Page 9, lines 352-353: Please replace here physical activity with PA

RESPONSE: It is now replaced.

  • Page 9, line 359: Please add a dot (.) after [39].

RESPONSE: It is now added.

References

  • Regarding reference section, please follow the format style of the Children’s journal (MDPI).

RESPONSE: Amended accordingly.

  • In many references, the doi is missing. Please enter the dois.ž

RESPONSE: DOI is added for all references

  • In many references (e.g., refs’ number 9, 14, 18, 19, 26), the dois are not in line - order with the other references (they protrude - stick out of the line). Please correct them.

RESPONSE: Corrected, thank you!

  • Page 11, line 427: Promotion—An – Please correct this reference here.

RESPONSE: It is now corrected.

Thank you once again! 

Staying at your disposal!

Authors

Reviewer 2 Report

Title:

Evaluation of physical literacy in 9- to 11-year-old children: re- 2 liability and validity of two measurement tools in three south- 3 eastern European countries

The reviewer’s comments

The subject matter of this theme is good and well worth pursuing. However, the reviewer would like to see some revisions made to your manuscript.

1. The abstract will be revised to include details pertaining to various aspects such as data collection, participants, methodologies employed, and the instruments utilised.

2. The introduction should include problem context, literature review and the hypothesis based on the gap analysis of the previously published research. Significance of the study should be elaborated in further depth. Justify

3. A detailed critique of recent studies should be in the Literature Review.

4. Methodology is well elaborated and detailed. However please elaborate the theory or framework used to carry out particular enquiries.

5. Conclusion is elaborated as per requirement.

6. Revisions or explanations are all made according to the suggestions of the reviewer. Requires Major Revision.

Title:

Evaluation of physical literacy in 9- to 11-year-old children: re- 2 liability and validity of two measurement tools in three south- 3 eastern European countries

The reviewer’s comments

The subject matter of this theme is good and well worth pursuing. However, the reviewer would like to see some revisions made to your manuscript.

1. The abstract will be revised to include details pertaining to various aspects such as data collection, participants, methodologies employed, and the instruments utilised.

2. The introduction should include problem context, literature review and the hypothesis based on the gap analysis of the previously published research. Significance of the study should be elaborated in further depth. Justify

3. A detailed critique of recent studies should be in the Literature Review.

4. Methodology is well elaborated and detailed. However please elaborate the theory or framework used to carry out particular enquiries.

5. Conclusion is elaborated as per requirement.

6. Revisions or explanations are all made according to the suggestions of the reviewer. Requires Major Revision.

Author Response

2ND REVIEWER

Comments and Suggestions for Authors

Title:

Evaluation of physical literacy in 9- to 11-year-old children: re- 2 liability and validity of two measurement tools in three south eastern European countries

The reviewer’s comments

The subject matter of this theme is good and well worth pursuing. However, the reviewer would like to see some revisions made to your manuscript.

RESPONSE: Thank you for recognizing the potential our manuscript and for giving us the opportunity to improve it. We amended the manuscript according to your suggestions. Please see in the forthcoming text how we dealt with each of your comments and where to find the changes. Staying at your disposal!

  1. The abstract will be revised to include details pertaining to various aspects such as data collection, participants, methodologies employed, and the instruments utilised.

RESPONSE: Abstract is amended and all necessary details are added. Please see changed text.

  1. The introduction should include problem context, literature review and the hypothesis based on the gap analysis of the previously published research. Significance of the study should be elaborated in further depth. Justify

RESPONSE: We tried to justify the study in more details, and text now reads: “However, the reliability and validity of the PL-measurement tools in southeastern Europe is rarely evidenced, and only one study examined this issue in adolescents, while: (i) there is an evident lack of research examining this issue in younger children, and (ii) to the best of our knowledge no study has been conducted on elementary school children in south-eastern Europe (the territory where similar Slavic languages are spoken).” (please see last paragraph of the Introduction)

  1. A detailed critique of recent studies should be in the Literature Review.

RESPONSE: In the introduction we tried to cover all necessary aspects of the study and provided critique of recent studies. However, the most important issue was that PL in elementary school children was not studied so far in the region, and therefore we focused on three countries where similar languages are spoken.

  1. Methodology is well elaborated and detailed. However please elaborate the theory or framework used to carry out particular enquiries.

RESPONSE: Thank you for recognizing the quality of our methodology. In this version we additionally extended it by calculating effect size differences for significant t-test results (please see Statistics and Results, thank you).

  1. Conclusion is elaborated as per requirement.

RESPONSE: Thank you.

  1. Revisions or explanations are all made according to the suggestions of the reviewer. Requires Major Revision.

RESPONSE: Once again, thank you for your comments and suggestions. We tried to cover it and amended manuscript accordingly.  

Staying at your disposal!

Authors